# S Phase Duration Is Determined by Local Rate and Global Organization of Replication

**DOI:** 10.3390/biology11050718

**Published:** 2022-05-07

**Authors:** Avraham Greenberg, Itamar Simon

**Affiliations:** Department of Microbiology and Molecular Genetics, IMRIC, Faculty of Medicine, Hebrew University of Jerusalem, Jerusalem 9112001, Israel; avraham.greenberg@mail.huji.ac.il

**Keywords:** cell cycle, S phase, DNA replication, replication timing, fork rate, inter-origin distance

## Abstract

**Simple Summary:**

In order for a cell to divide into two cells, it must first copy its DNA. Although the time required for this process tends not to vary much, many examples of the importance of variability have been reported. In this review, we discuss the methods used to study this question, present some of the examples of variation, and attempt to explain the factors that determine the time required in simple terms. We will show that the overall time depends on the rate of DNA replication within a region, and on the temporal organization of the regions relative to each other.

**Abstract:**

The duration of the cell cycle has been extensively studied and a wide degree of variability exists between cells, tissues and organisms. However, the duration of S phase has often been neglected, due to the false assumption that S phase duration is relatively constant. In this paper, we describe the methodologies to measure S phase duration, summarize the existing knowledge about its variability and discuss the key factors that control it. The local rate of replication (LRR), which is a combination of fork rate (FR) and inter-origin distance (IOD), has a limited influence on S phase duration, partially due to the compensation between FR and IOD. On the other hand, the organization of the replication program, specifically the amount of replication domains that fire simultaneously and the degree of overlap between the firing of distinct replication timing domains, is the main determinant of S phase duration. We use these principles to explain the variation in S phase length in different tissues and conditions.

## 1. Introduction

In order for a cell to pass its genetic material on to its progeny, it must first make a complete copy. This process, the synthesis of DNA, is highly regulated in order to ensure that it is successfully completed every generation. It is customary to divide the cell cycle into the following four phases: the pre-DNA synthesis phase (G1), the synthesis phase (S), the post-DNA synthesis phase (G2), and mitosis (M) [1,2]. In bacteria, G1, S and G2 are known as B, C and D periods, respectively. Although the order in which the genome is replicated has been studied in intricate detail (for example, see reviews [3,4,5,6,7]), the duration of replication has only recently been appreciated as important in cell development and fate (see Section 2.2). In this review, we will aim to take a step back and discuss how we can study the overall duration of S phase, what is known about this metric, and the factors that determine and regulate it. We will propose a simple model to explain how the duration of S phase is determined by the rate of replication of individual replicons, and by the temporal organization of replicon families.

## 2. Main

### 2.1. Measuring S Phase Duration

Measuring the total duration of a cell cycle is relatively easy, and can rely on the increase in cell number per time [8,9,10,11], coupled with methods to quantify the proliferative fraction, such as Ki67 labeling [12,13]. However, measuring the duration of individual stages of the cell cycle is much more challenging.

A rough estimation of the duration of each cell cycle stage can be achieved by FACS-based DNA content measurements that measure the fraction of the cells in each cell cycle stage. By measuring the amount of DNA in each cell of a population, it is possible to determine what percentage of the cells have 2N DNA content (G1), 4N (G2/M), or somewhere in between (S phase). In unsynchronized culture this is indicative of the relative duration of each stage (Figure 1A). The limitation of this simple and powerful technique is that it gives only relative results, namely one can determine by FACS what proportion of the cell cycle each stage lasts (without distinguishing between the G2 and the M phases) but with no independent knowledge of the duration of the entire cell cycle, it cannot be translated into actual units of time.

In bacteria this method is difficult to use, as bacteria can start a new round of replication before the previous round has completed [15]. However, by synchronizing the culture, preferably through a non-chemical method, and releasing cells into media with labelled nucleotides, it is possible to track the rate of synthesis of DNA and follow when a new round of replication has started. This makes it possible to calculate the time required for one round.

In order to simultaneously measure the proportion of cells in S phase and the absolute cell cycle time, several techniques were developed based on the incorporation of modified nucleotides (usually BrdU or EdU, or in the past, tritiated thymidine) during replication.

The cumulative labelling method (CLM) uses increasing exposure times to the modified nucleotide and measures the changes in the proportion of labelled cells over time (Figure 1B) [14,16,17,18]. The initial proportion of labelled cells indicates the percentage of cells in S phase at any given time, and the rate of increase of labelled cells allows the calculation of total cell cycle time (which can then be used to find S phase duration). A similar approach, also based on measuring the rate at which cells enter S phase, uses consecutive pulses of EdU and BrdU in order to determine the fraction of single-labelled cells. The fraction of cells labelled only with the second color can be used to calculate S phase duration (Figure 1C). These single-stained cells entered S phase during the time period of the second pulse, and they show what proportion of S phase duration is occupied by the second pulse. This in turn makes it possible to calculate the S phase duration. The advantage of this method over the CLM method is that it allows S phase duration to be measured in a single experiment and does not require conducting measurements over multiple time points. However, a key limitation of this method is the assumption that the sensitivity of incorporation and detection of both analogues is similar. If one analogue is preferentially detected, one can no longer rely on the fraction of single-labelled cells to determine S phase duration. In addition, a single time-point method cannot detect whether the population is uniform or composed of subpopulations with distinct cell cycle features. In the CLM method, which measures the accumulation of cells in S phase at multiple time points, a mixture of two populations will be seen as a different rate of S phase cell accumulation and a change in the slope of the graph.

Other methods are based on measuring the rate of exit from S phase. This can be done by pulse labelling cells with BrdU followed by a chase of increasing length. Originally, the exit from S phase was measured by counting the percentage of labelled mitotic (PLM) cells, which allows the determination of the length of G1, G2 and S phases [19,20,21,22] (Figure 1D). Briefly, cells are observed under a microscope and mitotic cells are identified. For each timepoint (chase-length), the percentage of the mitotic cells who have incorporated the signal is calculated. By noting the minimum and maximum chase length for mitotic labelled cells, the duration of each phase is calculated. Later on, this method was adapted for FACS, using BrdU antibody as well as a quantitative DNA stain such as PI or DAPI, to determine the percentage of S phase cells that reach G2 and G1 phases. It is also possible to add a second pulse after the chase to identify the length of time after which cells labelled with the first pulse are no longer labelled with the second (Figure 1E) [23]. A variety of formula have been published to calculate S phase duration from pulse-chase FACS data [24,25,26], and it has been adapted to also work as a single-sample assay [27,28]. However, this methodology is not robust and is very sensitive to the model used.

Another approach is based on EdU intensity, measured by flow cytometry, of cells that were pulse labelled with EdU for increasing durations. EdU intensity increases only for pulses shorter than the length of S phase, and reaches saturation when the pulse length is equal to the length of S phase [29] (Figure 1F). By conducting simple analysis of the FACS data it is also possible to calculate the time spent in G1 and in G2/M. Although this method worked well in its proof of concept, it is very sensitive to accurate and quantitative measurements of EdU intensity, which is not always possible, and it has not been widely used in recent publications.

A different approach for measuring the duration of cell cycle phases is based on live-cell imaging of cells labelled differentially in the various cell cycle phases. The FUCCI system, which genetically fuses fluorescent markers to two different proteins whose expressions oscillate inversely throughout the cell-cycle, allows measuring G1 and S/G2 durations [30]). In order to be able to differentiate between S and G2, another marker was added to the system (PCNA) that allows the detection of S phase cells [31,32]. These methods are based on live cell imaging of cells grown under the microscope for long times, conditions which can very possibly affect cell cycle parameters. Moreover, it requires the generation of specific cells carrying the labelled proteins and thus cannot be conducted in every system. Yet, it gives the advantage of being able to measure variability within the population and not only the average durations.

All of these methods are generally applicable and in principle should work in any type of organism. There are, however, some requirements that have to be met: first, BrdU-based methods can only work in organisms that can incorporate BrdU. Secondly, the culture must be unsynchronized in order for the fraction of labelled cells to indicate the population’s cell cycle dynamics. Finally, most of the methods will work only if the cell have similar S phase durations and for some methods also similar DNA contents.

### 2.2. Variability in Duration of S Phase

For over 50 years there have been reports of variability in S phase duration between cells [33,34], but these used to be thought of as anomalies and the S phase duration was broadly viewed as constant [35,36]. Yet, accumulating information suggests that there is a high degree of variability in S phase duration (Appendix A). Recently it has been proposed that variability in S phase duration may play a regulatory role and be associated with cell fate transitions, with shortened S phases associated with changes in cells fate such as differentiation, whereas longer S phases are correlated with stem cell maintenance (reviewed in [37]). This realization has mirrored the understanding that phenotype is not just a result of the linear DNA sequence, but involves multiple layers of regulation. This has led to the understanding that the role of S phase is not just to replicate the genome, but also to package it appropriately. DNA replicated at different stages of S phase has different chromatin structure which would indicate that the passage of a cell through S phase can affect and reflect its functionality.

Different organisms differ in the time it takes to replicate their genomes. Available data indicates significant variation between phyla and between developmental stages. Appendix A includes representative data for a range of bacteria, archaea, plants, metazoan and diatoms.

Many metazoans undergo a series of very rapid cell divisions shortly after the embryo is fertilized, requiring startlingly quick replication of their genomes. The quickest recorded genome replication is found in early *Drosophila* embryos, where the entire 240 Mbp of DNA is replicated in around 3–4 min [38]. *Xenopus laevis* embryos divide on average once every 35 min for the first 12 divisions [39] and a similar phenomenon has been seen in a variety of organisms including *Bufo cognatus* [40], *Hydractina* [41], and zebrafish [42]. Relatively rapid embryonic cell cycles can be found in all phyla [43] including in mammals, which complete their entire cell cycle in around 2.2-h in early embryonic mice and rats (6.5 and 8.5 days, respectively) [44,45].

In animals, after the rapid embryonic divisions most organisms tend to have much slower S phases. For example, the frog cell line A-6 has an S phase of 13.8 h at 28 °C [46], and the *Drosophila* Kc cell line requires 10 h, a stark contrast with the earlier rapid cycles. The fastest somatic mammalian cells which we have seen reported are activated CD8+ T cells who have S phases of under 5 h ex vivo [47] and in vivo CD8+ T cells can complete their whole cell cycle in under 2 h [48]. The vast majority of mammalian S phases are around 8–10 h (see Appendix A).

### 2.3. Genome Size

A naïve yet appealing assumption would be that S phase duration is associated with the size of the genome. Intuitively, the more DNA present in a cell, the longer the synthesis should take. Indeed, initially, a correlation between genome size and S duration has been shown in a variety of plants [49,50,51], though other studies failed to find a clear correlation [52,53,54,55]. It is possible that the association with genome size is found only in organisms including *Allium* and *Tradescantia*, with massive genomes of tens of Gbp [53]. Regardless, genome-size cannot be the sole determinant of S phase duration, as despite containing 200 times more DNA, human S phases are only approximately 35 times longer than those of budding yeast. Similarly, the differences in S phase duration between tissues in the same organism (see previous paragraph), demonstrates the involvement of other factors, other than genome size, in determining S phase duration. We have attempted to collate a wide variety of known measured S phase durations (see Appendix A. which is certainly not exhaustive, but aims to be representative of all organisms) and it appears that broadly speaking there is no correlation, with the exception of some of the very large plant genomes. It is possible that in the extremely large genomes the mechanisms usually employed to maintain S duration (such as organization of families of replicons, discussed below) are insufficient.

It would seem that we could easily resolve the question of whether S phase duration is correlated with genome size by looking at polyploid models. An easy example of this is in plants where it is possible to compare different ploidy within a species. In *Zea mays* roots it has been demonstrated that diploid and tetraploid cells have the same S duration [56], and the same has been seen in other plants [57]. However, ploidy is not conclusive proof that genome size does not affect duration of S phase because if the DNA content is doubled, there will be twice as many copies of the genes for replication factors and therefore replication factor concentrations will increase (assuming equal activity of genes in different ploidy nuclei). Indeed, human and monkey haploid and diploid embryonic stem cells have similar S phase durations [58] (Nissim Benvenisty, personal communication). Accordingly, one would expect that aneuploid cells will have shorter or longer S phase duration due to imbalance between replication resources and genome size. This helps explain the increased heterogeneity in S phase duration in aneuploid yeast cells [59].

### 2.4. Rate of Local Replication and Replication Organization

In order to consider the factors which determine the time taken to replicate a genome, we find it helpful to use the analogy of a simplified construction of a new neighborhood. For the sake of simplicity, we will consider the following parameters (with their biological counterparts in parentheses): the number of bricklayers per house (the density of replication origins), the speed of each bricklayer (replication fork rate), the availability of bricks (availability of replication factors), the subdivision of the neighborhood into smaller areas (the number of replicon temporal families), and the presence or absence of worker strikes (replication checkpoints). The total time required is dependent on two principal factors: how long it takes to build each house (or the rate of house buildings), and the average number of houses being built simultaneously. Each of these two considerations depends on two further things. The house building rate is determined by the number of bricklayers working on the house, and how fast each of them is working.

The average number of houses being built simultaneously is determined by how many areas the neighborhood is divided into, and also by how staggered the starts of these areas are from each other. The more areas we divide the neighborhood into, the fewer resources (e.g., builders and bricks) the whole neighborhood will need at any given time. For example, if there were no subdivisions at all, and all of the houses were considered to be one area, all of the houses will be started at the same time and the neighborhood will be completed in the minimum time. If, however, there are three projects of houses, and only one is being worked on at any given time, the neighborhood will require much longer to be built. It is clear that if all the workers were told to strike for a day, the time taken for the entire neighborhood to be built would increase.

In this analogy, the rate of house building represents the local replication rate (LRR), or 1/the time required to replicate one replicon. We normally measure this as an average of multiple genomic regions in multiple cells, by using DNA combing techniques. This gives us the replication fork rate (FR) which is equivalent to how fast the builder is working, and the inter-origin distance (IOD) which represents the number of builders working on one house (or the number of origins within a domain; Figure 2A). The average number of houses being built is the number of active replication forks genome wide (and not just within one domain). This is determined by the number of replicon temporal ‘families’ the genome is divided into, and the extent to which these families are staggered or overlap (Figure 2B). Finally, the worker strike is analogous to a pause in DNA replication due to the activation of the intra-S check point.

It is interesting to note that, in our house building example, the number of houses in the neighborhood has no effect. If you add more houses to the pre-existing housing projects, the neighborhood will still be completed in the same amount of time, provided that the house-to-builder ratio is maintained. However, if you were building just one house and only had one builder to do it, then the required time would depend on the size of the single house, and the speed at which the builder worked. This matches our discussion in Section 2.3 regarding the lack of correlation between genome size and S phase duration.

In bacteria the entire genome is replicated by a single origin [60] (for a review of the number of origins in archaea, see [61]) and therefore the LRR is determined only by the fork rate. In addition, with only a single origin there cannot be multiple replicon families, and therefore the organization of active replication forks cannot determine the duration of the period required to synthesize DNA. Accordingly, the time needed to complete the replication in bacteria can only be a function of FR and of the size of the genome. Indeed, it has been shown that changing nucleotide levels can modulate both prokaryotic FR and C period [62]. We are not aware of any studies comparing the time taken to replicate one copy of the bacterial genome with the genome size, but we assume that these would be highly correlated, and any discrepancies must be explainable by the LRR, which in turn depends solely on FR.

In eukaryotes the situation is much more complex. Because there are multiple replicons, the LRR is affected not only by speed (FR) but also by the number of forks simultaneously active. The contribution of replication organization to S phase length depends on both the length of the longest replicons, and on the timing of activation of the different origins. Although we tend to use measurements for FR and IOD which measure the averages, some replicons might be more responsible for the duration of S phase than others.

With this in mind, we will examine each of these factors and summarize known regulators of each.

### 2.5. Eukaryotic Local Replication Rate Depends on FR and the Number of Active Origins

The overall amount of DNA synthetized per unit of time (also known as replication capacity or replication potency) depends on the fork rate (FR), and the number of active origins (Figure 2A). While measuring FR is simple using the DNA combing methodology, the estimation of the number of active origins is more complex. In recent years it has become possible to use super-resolution live microscopy to visualize individual replicons in real time [63]. By multiplying replication foci numbers by intensity, it is possible to directly measure the amount of replication in a domain, at any given moment, which could previously only be inferred. An indirect way to infer the number of active origins is to use DNA combing to measure the average IOD [64]. A decrease in the IOD means that more origins are active in a region, yet it fails to measure what is happening at that time in other regions of the genome.

Restricted replication resources limit both LRR and the total number of active replicons. In other words, the raw materials can limit the amount of building per house, but also the amount of building in the neighborhood as a whole.

Interestingly, the LRR of the cells seems to be robust to changes in either FR or IOD, since FR and IOD are usually coupled and can compensate for each other [65,66,67,68,69]. Decreasing FR, for example by decreasing the nucleotide levels with hydroxyurea (HU) [70,71,72] or by interfering with DNA polymerase activity with aphidicolin [73,74,75], is followed by a decrease in IOD. The opposite is also true—interfering with origin firing, for example by depleting Cdc7 or Orc1 in yeast, causes an increase in FR [76]. Similarly, in an osteosarcoma cell line, both FR and IOD increased by approximately 50% as a consequence of knocking down various proteins involved in origin firing, such as Treslin [77]. Inhibiting the checkpoint protein ATR causes a decrease in FR [69] that is thought to be a consequence of the activation of additional origins, further demonstrating the coupling between origin density and FR. The change of IOD as a consequence of modifying the FR can be explained as a passive mechanism—a slowed FR gives more time for dormant origins to be activated [67], while faster forks reduce the likelihood of activation of neighboring origins due to their passive replication by the ongoing fork [65,68,76,78,79]. On the other hand, the opposite association, namely the effect of changing the IOD on FR, cannot be explained by a passive mechanism, which suggests that there are cellular mechanisms to ensure LRR homeostasis.

### 2.6. Fork Rate

The actual FR depends on several factors. (i) the intrinsic rate of the polymerase; (ii) concentration of nucleotides and other replication factors; (iii) chromatin that needs to be evacuated by the fork; (iv) obstacles (such as DNA damage and transcription machinery) that slow down the fork, pause it or even cause fork collapse. We will summarize what is known about each of these components.

The intrinsic replication processivity of the polymerase varies for different polymerases (reviewed in [80]), can be affected by mutations [81,82] and by aphidicolin, a drug that interferes with B-family DNA polymerases activity by competing with their ability to incorporate nucleotides [75,83]. In addition, it can be affected by temperature as has been seen in a variety of organisms [84,85,86].

DNA replication consumes dNTPs and therefore its rate depends on a constant supply of nucleotides [87]. dNTP levels increase by a factor of 3–6 in S phase compared to G1 [88], yet they are still limiting and at any given moment, the nucleotide pool is sufficient to replicate only 2% of the genome [89]. Thus, during replication a constant nucleotide supply, facilitated by RNR (ribonucleotide reductase) activity, is required to maintain the required dNTPs levels [90]. In normal, unperturbed S phase, ATR activity is responsible for maintaining dNTPs. Cells do not sense nucleotide levels prior to S phase, and instead they use ATR pathways (activated by nucleotide imbalance even without significant DNA damage) to increase nucleotide levels during S phase [91,92] as a result of the imbalance caused by the start of DNA synthesis [90]. HU is an RNR repressor, which reduces dNTP levels consequently reducing fork speed [70], which can be rescued by supplementing the cells with dNTPs. In addition, perturbations in dNTP pool levels have been achieved by RNR mutations [90,93,94], checkpoint activation [95,96]), and by artificially adding nucleotides [78].

The FR can be regulated by additional replication factors asides from dNTPs. Another factor which has been implicated as being key in controlling LRR by regulating FR is levels of the E2F family of transcription factors [97]. E2F-dependent transcription is known to upregulate transcription of replication factors, and is a key factor in the regulation of nucleotide concentrations [98].

Other known regulators of FR include the licensing protein MCM2 [99], ISG15 (known to regulate the RECQ1 helicase [100]), the PrimPol polymerase [79], Mrc1 (as discussed below regarding organization), the SUMO deubiquitinase USP7 and the p53/p21 pathway [77].

The FR can be increased by knocking down proteins associated with firing of origins, such as Treslin and MTBP, or proteins involved in the regulation of replication such as LIG1, FEN1, and DNA damage response like p21 or PARPi (e.g., Olaparib) [77]. The same is true of depletion of Orc1 and Cdc7 in yeast, both involved in origin activation [76]. In these cases, it is thought that the FR increases through an indirect increase in restrictive replication factors (less replication allows higher dNTP concentrations for the remaining forks).

Eukaryotic DNA is packed into chromatin and the replication fork must evacuate the histones before replication and allow time for their reassembly following replication [101]. In contrast, prokaryotes lack histones and this has been suggested to be the reason for the large difference between eukaryotic and prokaryotic FR (e.g., prokaryotes 20–60 kb/min; eukaryotes 1–3 kb/min [102,103]). In yeast, knocking down the acetyltransferase Rtt109 causes FR acceleration, probably by affecting nucleosome reassembly post-replication [104,105]. Live cell microscopy was used to directly measure the replisome progression in individual yeast cells and it was seen that the Winged Helix Domain (WHD) of the Cac1 histone chaperone complex is important for maintaining FR [106]. In addition, a number of proteins involved with chromatin transcription and nucleosome remodeling have been implicated in modulation of FR [101].

DNA damage causes stalling and in severe cases, collapse, of the replication fork, thereby affecting the actual speed at which the replication proceeds. Indeed, many factors that cause DNA damage affect FR in a checkpoint-independent manner [107]. Interestingly, HU affects FR not only through the dNTP pool but also by its effect on the reactive oxygen species (ROS) balance which causes DNA damage (reviewed in [108]). ROS are found to be elevated in cancer cells [109], and this leads to slower fork progression [110].

The checkpoint protein ATR is also known to have both a role in regulating nucleotide balance, and in limiting DNA damage by activating the intra-S checkpoint [76], both of which can affect FR [65,111].

Asides from the perturbations of FR which we have listed, there are various reports of natural regulation of FR. It has been shown that as cells progress to become pluripotent the FR increases, possibly through regulation of the level of available replication factors [112]. Similarly, the FR may change along S phase. Early studies showed a 3-fold increase in FR at late S phase compared with early S phase in mammals [113,114] and plants [115], however more recent findings have shown no such trend [116].

In addition to its role in establishing replication capacity, it is possible that fork rate can serve as a regulator of cell fate even where it does not change S phase duration [112]. It was shown that the slowing down of FR through a number of different techniques, stimulated embryonic stem cells to become totipotent. The slow FR and the higher density of origins was also accompanied by replication timing (RT) changes in specific regions of the genome.

### 2.7. Inter-Origin Distance (IOD)

IOD reflects the replicon size and can be measured directly using DNA combing. The IOD can vary between species and tissues. In yeast the average IOD is around 38 kb, in Chinese hamster, *Drosophila* and vascular plant cells the average is around 65–75 kb [117,118,119]. A comparison of cancerous and normal human keratinocytes found mean IODs of 124 kb and 120 kb respectively [68], while HeLa cells have been reported to have an IOD of 188.7 kb [63]. Two mouse cell lines, fibroblasts and stem cells, have IODs of 136 kb and 139 kb, respectively [118]. The *Drosophila* early embryonic stage has a very short IOD of 3.4 kb, and a remarkably short S phase, as discussed below [38].

As mentioned above, there is a strong coupling between FR and IOD which makes it difficult to identify factors that only affect one of them. However, separate effects on IOD and FR can be explored by chemical inhibition of origin activity (through a Cdc7 kinase inhibitor) or of DNA synthesis (via the DNA polymerase inhibitor aphidicolin) [79].

Some of the proteins previously mentioned as regulators of FR are associated with the firing of origins. In these cases, it is thought that the main process which changes is the IOD, and the FR increases through an indirect increase in limiting replication factors (less replication allows higher concentrations for the remaining forks) [76]. For example, IOD has been shown to decrease when the Treslin binding partner MTBP, Orc1 or Cdc7 (the kinase sub-unit of DDK, vital for replication initiation) were knocked down.

### 2.8. Replication Organization

In eukaryotes in which there are multiple replicons, not all replicons fire simultaneously. We can simplify the situation and imagine that there is a finite number of families of replicons, and each family is activated at a different stage of S phase. Therefore, the length of S phase can be determined by how many of these families there are, and the degree of staggering between these families (Figure 2B). It is also the case that the activation of the intra-S checkpoint will extend S duration by increasing the gaps between activation of different replicon families.

The RT program has been studied at a variety of resolutions, using many different protocols, and is known to be a highly regulated, cell-type specific epigenetic marker [5]. Replicons are organized in clusters or domains) that fire simultaneously, separated by temporal transition regions (TTRs) that are most probably replicated by a single replication fork [120,121]. However, the relative order of different regions of the genome does not determine the duration of S phase. To return to our analogy, if the developers decided to swap the order in which the various areas are built, this does not need to affect the overall time required. In addition, it is not trivial to detect changes in the staggering of origins using existing RT data when the order of origin activation is unaffected.

### 2.9. Examples of S Phase Duration Regulation

By studying examples where S phase length changes, we can try and identify the relative contributions of LRR, and staggering/replicon families.

In most known cases where S phase duration changes, this is not achieved by a change to the LRR, which is generally robust to changes, often due to the effects that the IOD and FR have on each other, as discussed above (although the compensation is not always sufficient to fully preserve S phase duration [23]). In addition, the RT program does not normally change much despite variance in the duration of S phase. By a process of elimination, it is often possible to conclude that the main factor responsible for determining S phase duration is the staggering of activation of domains, or a change in the number of replicon families. In addition, pausing replication by activating the mid-S checkpoint will lengthen S phase. It should be noted that both overlap between families and changing the number of replicon families depends on the presence of sufficient nucleotides and other replication factors.

The dramatic change in S phase duration during midblastula transition (MBT) in which cells transit from an extremely short S phase stage (termed cleavage cycles) to regular length S phase have been extensively studied. Those changes are probably a consequence of a change in the DNA/cytoplasm ratio [39], since there is no expression of new proteins yet at this stage, which changes the availability of key replication factors. Both LRR and replication organization play a role in these dramatic changes in S phase duration. The early embryonic divisions exhibit very short IODs, and the origins are much more regularly spaced than somatic origins [122]. The regular spacing of origins, especially in *Drosophila* where there is an origin every 3.4 kb [38], is thought to be instrumental in ensuring the short S phase. Similarly, the FR of early embryonic stages is 3 times faster than FR of somatic cells [123]. Yet, these changes in LRR cannot explain the dramatic difference in S phase duration alone. Indeed, replication organization is also changed at MBT by increased overlap of origin firing, with early and late replication domains firing almost simultaneously at the early embryo stages. The delay of late replicating origins is introduced at MBT, as a consequence of Rif1 activation by Cdk1 which causes a delay in the replication of heterochromatin. The decreased overlap appears to be in part achieved by titration of a few key replication factors, but not dNTPs [64], leading to the activation of the intra-S ATR/Chk1 checkpoint [124]. In addition, after *Xenopus* MBT, S phase duration increases despite no immediate change in local IOD [64].

The synthesis of DNA immediately prior to meiosis takes on average twice as long as mitotic synthesis in a number of organisms [125]. Evidence in yeast shows that the LRR in yeast premeiotic S phase is the same as the normal LRR, and that the RT program of origin activation is also similar [126], suggesting that the slow S is a consequence of decreased overlap of replicon family activation. However, there are organisms, such as the newt *Triturus cristatus*, in which the lengthening of the premeiotic S phase is probably in part a consequence of a decrease in the LRR, through increased IOD [127].

Another interesting example of S phase extension is found in some cancer cells. There have been reports that S phase in cancer cells tends to be around twice as long as that of normal cells [128]. A study which measured the overall global replication activity (or ‘potency’ in their words), found fewer simultaneously active forks. This explained the slower S phase despite an increase in FR [129].

In the meristem roots of vascular plants, it was shown that IOD and FR are not correlated with S length, and the S length is instead regulated through replicon family staggering [119], where the late origins are fired after a ‘tempo-pause’ [130]. The presence of a correlation between vascular plant S phase duration and genome size (see above) is due to relatively strict limits on the number of replicons simultaneously active in a plant nucleus. This means that more families have to be introduced to deal with larger genomes. If, however, in animal cells the families are not near full occupancy, there can be enough flexibility to ensure that there is no need to change the number of families or the extent of staggering even when dealing with large genomes.

Multiple genes associated with S phase duration have been found, most notably through a genome-wide screen for replication mutants. 14 yeast genes extended S phase duration when knocked down. These genes included genes related to cell-cycle progression, the DNA replication machinery, and nucleotide metabolism. Interestingly, it was shown that for most mutants the relative order of origin firing was maintained and the extension of S was due to coordinated delay in the activation time of all middle and late origins. This is a good example of the staggering of replication organization. By delaying the activation of later domains, the total amount of replication at a given time decreases, although not necessarily the local capacity. Strikingly, the scaling of the replication timing program was completely lost in mrc1 cells in which only a small fraction of origins, confined to the very end of S phase, were delayed, causing a two-fold extension in S phase duration [131]. In mammals, certain replication factors are known to be involved in controlling FR, as discussed above.

It has been suggested for 50 years that nucleotide availability could be a key regulator of S phase duration [132]. This has been shown chemically and genetically and is commonly explained by the effect of nucleotide concentrations on LRR. Overexpression of RNR increases nucleotide levels which can increase FR and in turn shortens S phase in bacteria [62], where the capacity is determined solely by FR, as discussed above. However, as mentioned above, changing FR in multi-origined lifeforms leads to compensation by IOD [65,66]. Therefore, we believe that a main pathway through which nucleotides could regulate S phase duration is by affecting the ability of the cell to replicate more origins simultaneously, whether by decreased overlap, or by decreased number of replicon families. In a case where the concentration of nucleotides, or other replication factors, is restricting the organization, an increase or decrease in concentration could result in a change to the length of S phase, as observed in budding yeast [78,90]. Indeed, it has been shown that increased dNTP levels can cause late origins to be activated early [91].

## 3. Conclusions and Open Questions

We have highlighted the importance of replication organization in determining S phase duration. This conclusion was achieved mainly through the elimination of other potential factors that may influence S phase length, rather than by direct measurement of replication organization. This is due to the lack of methods to measure the mesoscale organization of the replication program which has limited our knowledge about the number of replication domains that are simultaneously firing and the degree of overlap between them. Further study is needed to highlight this important aspect of the organization of the replication program. The ability to quantify the total amount of DNA replication at a given time, on a single-cell basis, might lead to an increased ability to explore the cases where the replication program appears unchanged despite a change in S phase duration. In addition, recent advances in the understanding of the 3D compartments of DNA might be useful in exploring the replicon families as a limiting factor in S phase duration. These compartments may be separated from the rest of the nucleus through a phase separation process and further research is needed to assess the contribution of phase separation to the organization of DNA replication and S phase duration. In the future it might also become more convenient to measure the concentrations of replication factors within individual replication foci which would increase the ability to separate between LRR and staggering.

## Figures and Tables

**Figure 1 biology-11-00718-f001:**
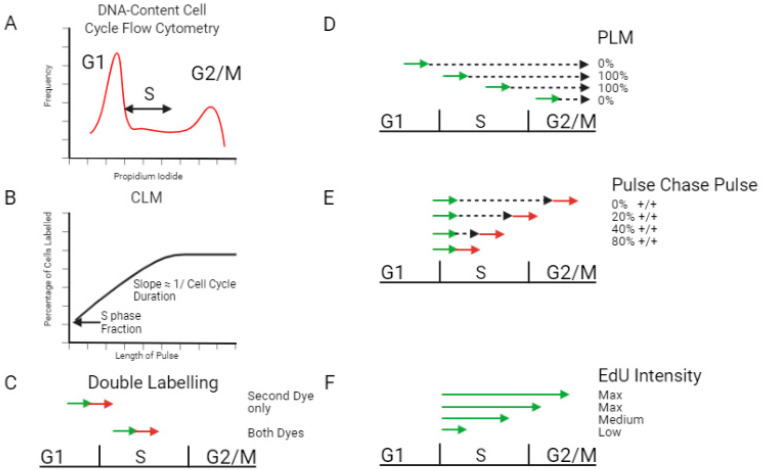
Methods to measure S phase duration in an unsynchronized population. (**A**) Flow Cytometry using a DNA-dye to Quantify the Amount of DNA Present. This allows the percentage of cells in S phase to be calculated. (**B**) Cumulative Labeling method. Cells are exposed to increasing pulses of labelled nucleotides. When 100% of cells are labelled, it can be deduced that the pulse length was equal to the duration of G1+G2+M. An approximation for the slope can be calculated following reference [14]. (**C**) Double Labeling. The proportion of cells labelled only with the second dye indicates rate of exit from S phase. (**D**) Percentage Labelled Mitosis. Cells are exposed to a short pulse of labelled nucleotides (green arrow) and then increasing chases (black dashed arrow). The percentage of mitotic cells containing labelled nucleotides is tracked over time. (**E**) Pulse Chase Pulse. Cells are exposed to short pulses of different colors (green and red arrows), with increasing chases in between (black dashed arrow). The percentage of S phase cells labelled with both dyes is shown (**F**) EdU Intensity. Cells are exposed to increasing pulses of EdU (green arrow). When maximal intensity is reached it indicates that a population of cells have been exposed to EdU for the entire length of S phase. Figure created with Biorender.com, accessed on 5 April 2022.

**Figure 2 biology-11-00718-f002:**
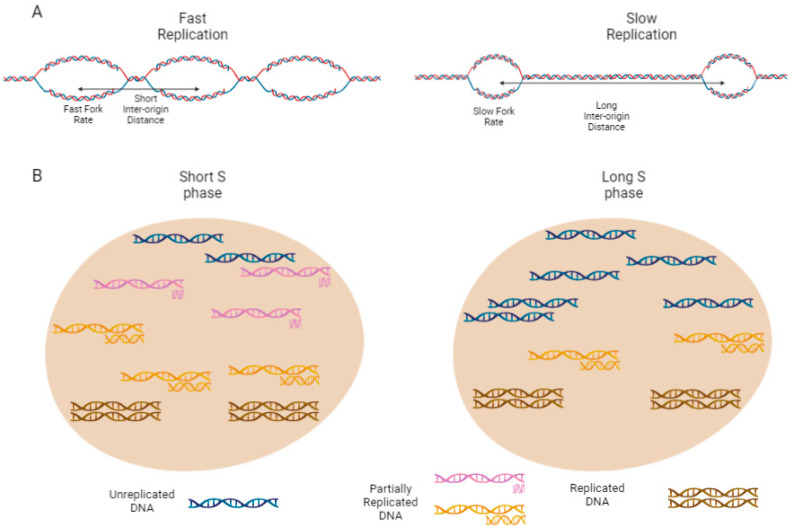
Illustration of factors which can potentially determine S phase duration. (**A**) Local replication rate (LRR). The left DNA replication domain has got a fast fork rate (hence, large replication bubbles), and a high origin density (and therefore short IOD). This means that it has a high LRR and will therefore be replicated quickly. The domain on the right has got a slow fork rate and low origin density. This means that it has a low LRR and will therefore take a long time to be replicated. (**B**) Global replication organization. Each nucleus is a snapshot of the middle of S phase. Each stretch of DNA represents one domain, and the legend and color code are explained at the bottom of the figure. In the left nucleus, the temporal families are big, with 3 domains in orange and 3 in pink. In addition, there is an overlap between the replication of the pink domains that start replicating before the completion of the replication of the orange domains. This demonstrates a short S phase. The nucleus on the right has no overlap of replication and also only has 2 domains per family, both of which mean that S phase will take longer. Figure created with Biorender.com, accessed on 5 April 2022.

## Data Availability

Not applicable.

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
