# Peer review of "S Phase Duration Is Determined by Local Rate and Global Organization of Replication"

_biology, 2022, doi:10.3390/biology11050718_

Round 1

Reviewer 1 Report

Title:  S phase duration is determined by local rate and global organization of replication 

In this review, Greenberg and Simon address how S phase length is controlled in different systems. The authors made a great effort in revising the methodology used in the field and the data available regarding S phase duration in many cells/organisms. The organization of the review is reasonable and the conclusions are in agreement with the revised literature.

However, the style of the writing makes the story a bit difficult to follow in some parts:

The first part, devoted to the different methods to measure S phase duration, would benefit from a more detailed and clear explanation of how the procedure works and how the results are interpreted.

Also, in particular in the part explained as “DNA replication as a building project”, the analogy does not help to understand the concept they are trying to put across and adds a layer of confusion to the whole thing. And Figure 2 does not help to clarify it either.

Minor points:

in lines 291-295 the authors write:

“While the decrease of IOD as a consequence of slow FR can be explained as a passive mechanism – a slowed FR gives longer time for dormant origins to be activated (suggested by [76]) – the opposite association cannot be explained by such mechanism which suggests that there are cellular mechanisms to ensure LRR homeostasis.”

However, it has been suggested that faster forks reduce the likelihood of activation of neighbouring origins due to its passive replication by the ongoing fork. (Anglana et al, 2003, Cell, 385; Zhong et al, 2013, JCB, 373; Conti et al, 2007, Genes Chormosome Cancer, 724; Poli et al, 2012, EMBO J, 883; Rodriguez-Acebes et al, 2018, JBC, 12855). Please, ammend.

in line 303 the authors write:

“…by aphidicolin, a drug 303 that interferes with DNA polymerase alpha activity by competing with its ability to incorporate dCMP”.

However, aphidicolin is an inhibitor of B-family polymerases, including Pold (Baranovskiy et al, 2014, NAR, 14013). Please, rephrase.

Finally, the manuscript would benefit from a simpler style and a more direct writing. Although the review is interesting, the manuscript is not easy to understand in its present form.

Reviewer 2 Report

This review provides a comprehensive and detailed explanation of factors that contribute towards the duration of S phase in different species and cell types. The cell cycle is divided into 4 phases (G1, S, G2, M) and a high degree of variability exists in the duration of cell cycle timing among different species. In this review, the authors highlight the importance of variability in S phase and describe various factors that contribute to this variability. Finally, they highlight the importance of replication program in determing the length of S phase.

General Comments: Overall, this review is very detailed.

  • For the benefit of the reader, it would be helpful if the authors could highlight some of voids in our current understanding of replication rate and how future studies might be able to address these voids.
  • In section 2.1: Measuring S phase duration: The authors have very nicely described all the different methods used to measure S phase, along with their limitations and advantages. In addition, to the information provided, it would be helpful if the authors could elaborate whether the suitability of methods varies with different species or cell types with specific examples from the literature.
  • It would be nice if the authors could include more illustrations describing the text.
  • The references cited by the authors are very old. Approx., 28% references are older than 30 years and can be replaced by newer ones, while only 33% references are from the last 10 years.

Round 2

Reviewer 1 Report

Title:  S phase duration is determined by local rate and global organization of replication 

In this review, Greenberg and Simon address how S phase length is controlled in different systems. The authors made a great effort in revising the methodology used in the field and the data available regarding S phase duration in many cells/organisms. The organization of the review is reasonable and the conclusions are in agreement with the revised literature.

Regarding my previous comments, the authors have revised the points I raised and re-written the manuscript accordingly.

The part where they explain the different methods to measure S phase duration has improved and is easier to understand.

And the item “Rate of Local Replication and Replication Organization” is now clearer and more comprehensible and the authors made a great effort to re-design Figure 2 to make the topic accessible to a wide audience.

Minor points:

The minor points have been addressed.